# Adherence to an overweight and obesity treatment: how to motivate a patient?

Isaac Kuzmar[1], Mercedes Rizo[1] and Ernesto Cortés-Castell[2]

[1] Faculty of Health Sciences, University of Alicante, Alicante, Spain
[2] School of Medicine, Miguel Hernández University, Alicante, Spain

## ABSTRACT

**Objective.** To explore anthropometric changes in normal-weight, overweight and obese subjects who did not dropout or fail a weight loss program over the 16 treatment weeks to improve patient motivation and treatment adherence.

**Methods.** A clinical intervention study was conducted among 271 (including 100 dropouts and/or failures) obese and overweight patients who consulted a nutrition clinic in Barranquilla (Colombia) for the purpose of nutritional assessment. They were subject to a personalized weekly follow-up consultation over the course of 16 weeks in which initial and the final Body Mass Index (BMI, kg/m$^2$), photographs, food consumption patterns, percentage weight loss, waist and hip circumference were registered and grouped according to BMI, measuring treatment response. Data's nonparametric statistical comparison was made.

**Results.** In 62 patients from the BMI < 25 group, there is weight loss of 2.6% (3.1 SD), 5.5% (3.3 SD) in waist circumference and 3.0% (2.5 SD) in hip circumference. In 67 patients from the $25 \geq$ BMI < 30 group, there is weight loss of 3.8% (4.1 SD), 5.7% (4.5 SD) in waist circumference loss and 3.7% (3.0 SD) in hip circumference loss. In 42 patients from the BMI > 30 group, there is weight loss of 4.8% (3.7 SD), 7.0% (3.6 SD) in waist circumference loss and 3.9% (2.4 SD) in hip circumference loss. Monitoring is done every 4 weeks by the Friedman test, with significant differences between the three groups ($p < 0.001$). Patients do not drop out of treatment because they start to see physical results in waist decrease. When comparing final values of initial waist/hip circumference ratios and waist/height ratios, a clear decrease in the three BMI groups was observed ($p < 0.001$).

**Conclusion.** After three weeks of continuous treatment patients improved in all overweight and obesity parameter indicators; there were not statistically significant differences in hip circumference (HC) and waist loss (WC) (%) among the three BMI groups (normal-weight, overweight, and obesity). In contrast, there were statistically significant differences in weight loss (%) and waist-to-hip ratios. Based on anthropometric outcomes and patient perception of their body image it can be concluded that the waist circumference loss is the parameter that retains obese patients in the weight loss program.

Corresponding author
Isaac Kuzmar,
isaackuzmar@yahoo.es

## INTRODUCTION

Overweight (body mass index, BMI 25–30 kg/m$^2$) and obesity (BMI $\geq$ 30 kg/m$^2$) are preventable diseases defined as abnormal or excessive fat accumulation that sometimes favours the onset of disease (*WHO, 2013*).

Over the years, the obesity prevalence significantly increases (*Ogden et al., 2006*) due to a decrease in caloric expenditure and increased energy consumption, resulting from poor diet and sedentary lifestyle (*Centers for Disease Control and Prevention, 2013*) coupled with hormone physiopathology implications such as leptin (*Sørensen, Echwald & Holm, 1996*) and ghrelin (*Hinney et al., 2013*).

There is a close relationship between waist circumference and cardiovascular risk in obesity (*Masiá et al., 1998*); concerned men and women commonly lose weight by consuming less fat but not fewer calories rather than practicing the recommended combination of hypocaloric diet associated with physical activity (*Serdula et al., 1999*; *Wadden, 1993*) to achieve permanent changes in lifestyle (*SIGN, 1996*; *NHS, 1997*) allowing better obesity control (*Benítez Guerrero et al., 2009*).

Some studies reference marital status (*Cano Garcinuño et al., 2010*), level of education (*Mazure et al., 2007*) and social class (*Da Veiga, Da Cunha & Sichieri, 2004*) with overweight and obesity, but it has been shown that these parameters are not considered influential factors in the successful outcome of a treatment (*Kuzmar, Cortés & Rizo, in press*).

In some cases appetite suppressants that increase anorexigenic neurotransmitters in the central nervous system (*Yanovski & Yanovski, 2002*) such as sibutramine (*National Institute for Clinical Excellence, 2001*) and orlistat have been used, but the suppressants were only continued if patients lost weight and maintained the weight loss without significant side effects (*Noël & Pugh, 2002*). Currently these drugs are suspended (*Hernández García, 2010*) and are under consumption alerts (*Heitmann Ghigliotto, 2010*).

Alternative overweight and obesity treatments are very popular but despite being widely used, they have not been shown to be safe and effective (*Allison et al., 2001*).

In morbid obesity (BMI $\geq$ 40 kg/m$^2$) sometimes lifestyle changes are not enough, (*National Institute for Clinical Excellence, 2002*) necessitating bariatric surgery to achieve effective weight loss (*Morales et al., 2011*).

In clinical practice it is important to predict nonabdominal, abdominal subcutaneous, and visceral fat in patients by measuring BMI and waist circumference independently (*Janssen et al., 2002*).

This study therefore seeks to determine which of the parameters monitored to improve body image and overweight treatment: BMI decrease, weight percentage and waist and hip circumference loss, could serve as patient motivation.

## MATERIAL AND METHODS

### Subjects

A clinical intervention study was conducted among 271 (233 women and 38 men) overweight and obese participants who consulted a nutrition clinic in Barranquilla

(Colombia) for the purpose of nutritional assessment. They were subject to a personalized weekly follow-up consultation over the course of 16 weeks. The inclusion criteria were voluntary assistance, patient desire to improve their aesthetic image, excluding those with chronic diseases such as diabetes, kidney failure, etc., since patients came for aesthetic reasons. This study does not consider patients who tried a diet to lose weight in the previous month or earlier, as this aspect to analyse the resistance/adherence to current treatment is not necessary. In turn, alcohol and tobacco consumption do not affect actual results. 171 (63.1%) overweight or obese patients according to the WHO classification (*WHO, 2013*) continued the study. The sample was composed of patients from 15 to 80 years of age collected over a period of 3 years.

The study was conducted according to Helsinki's rules obtaining all patients informed consent.

## Methods

As in previous studies (*Kokkinos et al., 1995*), we assume changes in a nutritional treatment can be seen in 16 continuous weeks. The study included a patient's complete medical record and a weekly WHO recommended medical-nutritional assessment (*OMS, 1995*) by obtaining height, weight, waist and hip circumference data, as well as its own comparison of their initial and final treatment body image through photographs for self-perception control. We used an eating habits questionnaire similar to the Dana-Farber Cancer Institute questionnaire (*Dana-Farber Cancer Institute*), asking about background and habits at home and work that may relate to the patient's health focusing on eating habits. We made the weekly low calorie diets WHO-based (*WHO*) according to the questionnaire response.

With the obtained data we calculate the initial and final BMI according to WHO (*WHO, 2013*; *OMS, 1995*) criteria, as well as weight, waist and hip loss percentages.

The data were analyzed using IBM SPSS Statistics version 22.0 software, checking the normality and comparative nonparametric statistics on data that did not show a normal distribution by Friedman's test. A significance level of $p < 0.05$ is considered. This study was approved by SEMI-Servicios Médicos Integrados of Barranquilla, Colombia.

## RESULTS

The 63.1% of patients with successful loss in all the studied variables were analysed. 36.9% of patients dropped out during the first three visits with no known medical reason or significant relationships to sex and BMI; we assumed that patients discontinued the treatment because they did not get the immediate results in waist loss they expected. Changes begin to be perceived from the fourth week as shown in figures. Table 1 shows that in 62 patients from the BMI < 25 group, there is weight loss of 2.6%(3.1 SD), 5.5%(3.3 SD) in waist circumference loss and 3.0%(2.5 SD) in the hip circumference loss. In 67 patients from the $25 \leq$ BMI < 30 group, there is weight loss of 3.8%(4.1 SD), 5.7%(4.5 SD) in waist circumference loss and 3.7%(3.0 SD) in the hip circumference loss. 42 patients from the BMI > 30 group, there is weight loss of 4.8%(3.7 SD), 7.0%(3.6 SD) in waist circumference loss and 3.9%(2.4 SD) in the hip circumference loss. There were statistical

**Table 1 Results of initial and final BMI, weight, waist and hip circumference, and percentage loss in BMI groups at 16 treatment weeks (mean, standard deviation and 95% confidence interval).**

| BMI (kg/m$^2$) | <25 | 25 ≥ BMI < 30 | >30 | $p$ (Kruskal-Wallis) |
|---|---|---|---|---|
| $n$ | 62 | 67 | 42 | |
| $i$ BMI mean (SD) | 23.1 (1.3) | 27.5 (1.5) | 32.9 (3.5) | <0.001 |
| (CI 95%) | (22.7–23.4) | (27.2–27.9) | (31.8–34.0) | |
| $f$ BMI mean (SD) | 22.5 (1.4) | 26.5 (1.7) | 31.3 (3.3) | <0.001 |
| (CI 95%) | (22.1–22.8) | (26.1–26.9) | (30.3–32.3) | |
| Paired test ($p$) | <0.001 | <0.001 | <0.001 | |
| $i$ BMI - $f$ BMI mean (SD) | 0.6 (0.7) | 1.0 (1.1) | 1.6 (1.3) | <0.001 |
| (CI 95%) | (0.4–0.8) | (0.8–1.3) | (1.2–2.0) | |
| Weight loss % mean (SD) | 2.6 (3.1) | 3.8 (4.1) | 4.8 (3.7) | <0.05 |
| (CI 95%) | (1.8–3.3) | (2.8–4.8) | (3.6–5.9) | |
| $i$ waist mean (SD) | 76.3 (5.6) | 86.7 (7.3) | 100.8 (11.4) | <0.001 |
| (CI 95%) | (74.9–77.7) | (85.0–88.5) | (97.3–104.4) | |
| $f$ waist mean (SD) | 72.1 (5.4) | 81.6 (6.5) | 93.7 (10.0) | <0.001 |
| (CI 95%) | (70.7–73.5) | (80.0–83.2) | (90.5–96.8) | |
| Paired test ($p$) | <0.001 | <0.001 | <0.001 | |
| Waist loss % mean (SD) | 5.5 (3.3) | 5.7 (4.5) | 7.0 (3.6) | ns |
| (CI 95%) | (4.6–6.3) | (4.6–6.8) | (5.9–8.1) | |
| $i$ hip mean (SD) | 96.4 (5.3) | 105.8 (5.5) | 115.3 (7.3) | <0.001 |
| (CI 95%) | (95.0–97.7) | (104.4–107.1) | (113.0–117.6) | |
| $f$ hip mean (SD) | 93.5 (5.5) | 101.9 (5.9) | 110.8 (7.5) | <0.001 |
| (CI 95%) | (92.1–94.9) | (100.5–103.4) | (108.5–113.1) | |
| Paired test ($p$) | <0.001 | <0.001 | <0.001 | |
| Hip loss % mean (SD) | 3.0 (2.5) | 3.7 (3.0) | 3.9 (2.4) | ns |
| (CI 95%) | (2.4–3.6) | (2.9–4.4) | (3.2–4.7) | |
| $i$ waist/$i$ hip ratio mean (SD) | 0.79 (0.06) | 0.82 (0.07) | 0.88 (0.09) | <0.001 |
| (CI 95%) | (0.78–0.81) | (0.80–0.84) | (0.85–0.90) | |
| $f$ waist/$f$ hip ratio mean (SD) | 0.77 (0.06) | 0.80 (0.06) | 0.85 (0.07) | <0.001 |
| (CI 95%) | (0.76–0.79) | (0.79–0.82) | (0.82–0.87) | |
| Paired test ($p$) | <0.001 | <0.001 | <0.001 | |
| $i$ waist/$i$ height ratio mean (SD) | 0.48 (0.04) | 0.53 (0.04) | 0.62 (0.06) | <0.001 |
| (CI 95%) | (0.47–0.49) | (0.52–0.54) | (0.60–0.64) | |
| $f$ waist/$f$ height ratio mean (SD) | 0.45 (0.03) | 0.50 (0.03) | 0.57 (0.06) | <0.001 |
| (CI 95%) | (0.45–0.46) | (0.49–0.51) | (0.56–0.59) | |
| Paired test ($p$) | <0.001 | <0.001 | <0.001 | |

significant differences in waist, hip circumference (HC), waist circumference (WC), waist/hip ratio and waist/height ratio shown in Table 1 paired test ($p < 0.001$). Friedman's test monitoring for weight (Fig. 1), waist (Fig. 2) and hip (Fig. 3) loss is done every 4 weeks, with significant differences between the three groups ($p < 0.001$). Comparing final values of initial waist/hip circumference ratios and waist/height ratios, a clear decrease in the three BMI groups was observed ($p < 0.001$) (Table 1). When comparing self-perception

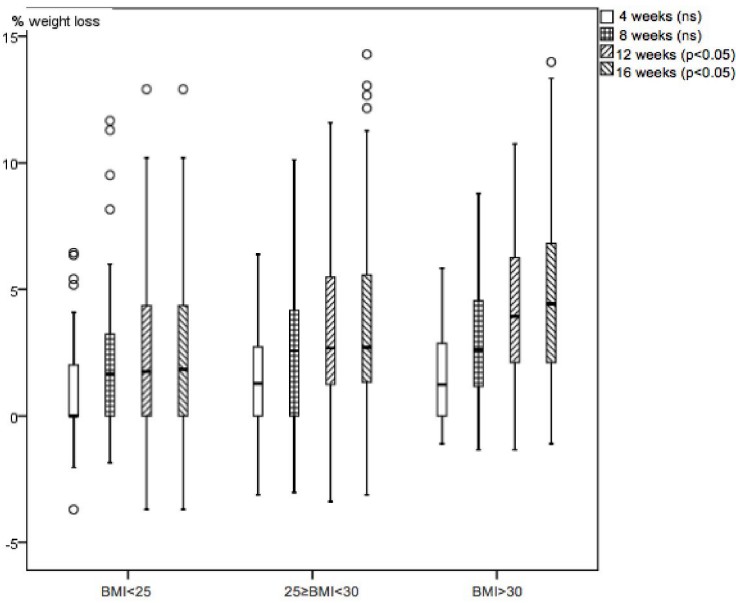

**Figure 1 Boxplot diagram.** Weight loss percentage monthly variation in the three studied nutrition groups. Friedman test $p < 0.001$.

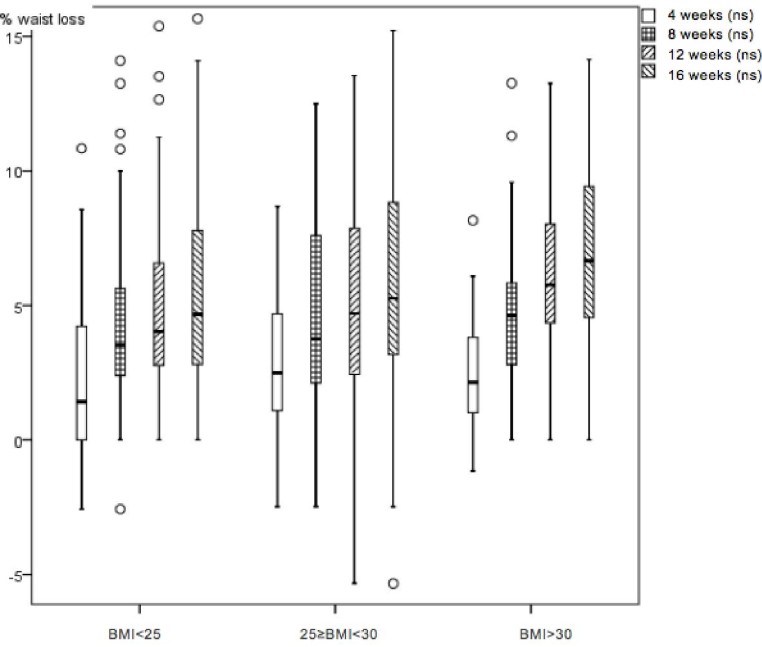

**Figure 2 Boxplot diagram.** Waist loss percentage monthly variation in the three studied nutrition groups. Friedman test $p < 0.001$.

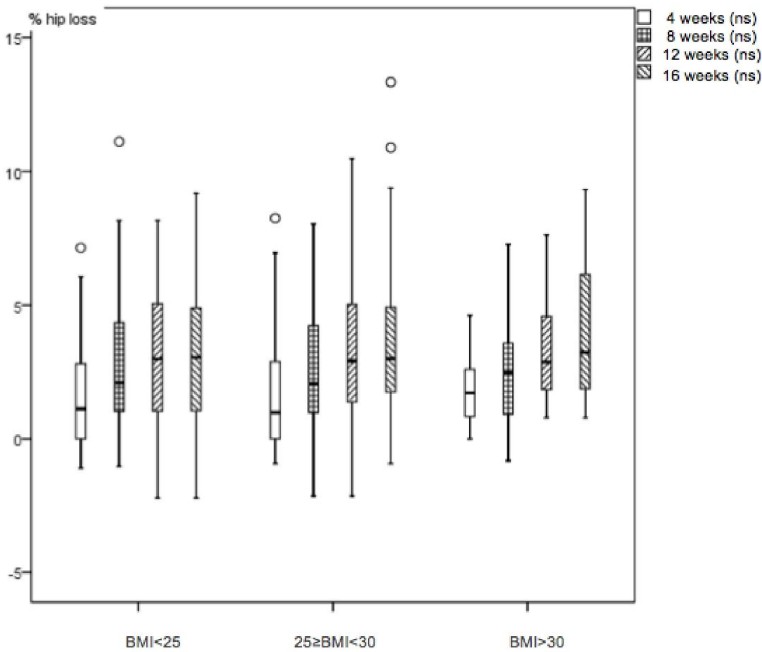

**Figure 3 Boxplot diagram.** Hip loss percentage monthly variation in the three studied nutrition groups. Friedman test $p < 0.001$.

data through the initial and final week patient treatment photos, they clearly showed satisfaction verifying their waist loss perception (Fig. 4).

## DISCUSSION

The concept of body image changes during life affecting individual behaviour (*Calado, Lameiras & Rodríguez, 2004*), so it cannot be separated from the weight loss. In the present study, we have obtained good results in a high percentage (63.1%) of patients who attended the consultation to improve their body image and/or weight loss. These success rates are highly variable in the literature (*Hill & Williams, 1998*; *Paxton et al., 1999*) and depends on many factors (*Paxton et al., 1991*).

It is observed that even patients who attended the consultation to improve their body image but were not overweight (*WHO, 2013*), lose BMI, weight, waist and hip, although it is noted that weight stabilizes after 8 treatment weeks. The overweight (*WHO, 2013*) group also stabilizes weight at 8 weeks; only the obese (*WHO, 2013*) group maintains an ongoing weight loss until the end of treatment and may indicate the need to extend it for more weeks.

In all cases, waist loss is superior to the other examined parameters. It continuously decreases for 16 weeks with greater decreases in the obese (*WHO, 2013*) group and doesn't plateau in any of the three groups. Thus, it is a parameter for which many patients seek superior tracking time, and is an appreciated body image index (*Casper et al., 1979*), with very visual and comparable-to-initial-state results (*Garner et al., 1980*). Hip losses are lower and temporarily appear similar to weight loss. Waist/hip ratio losses, after

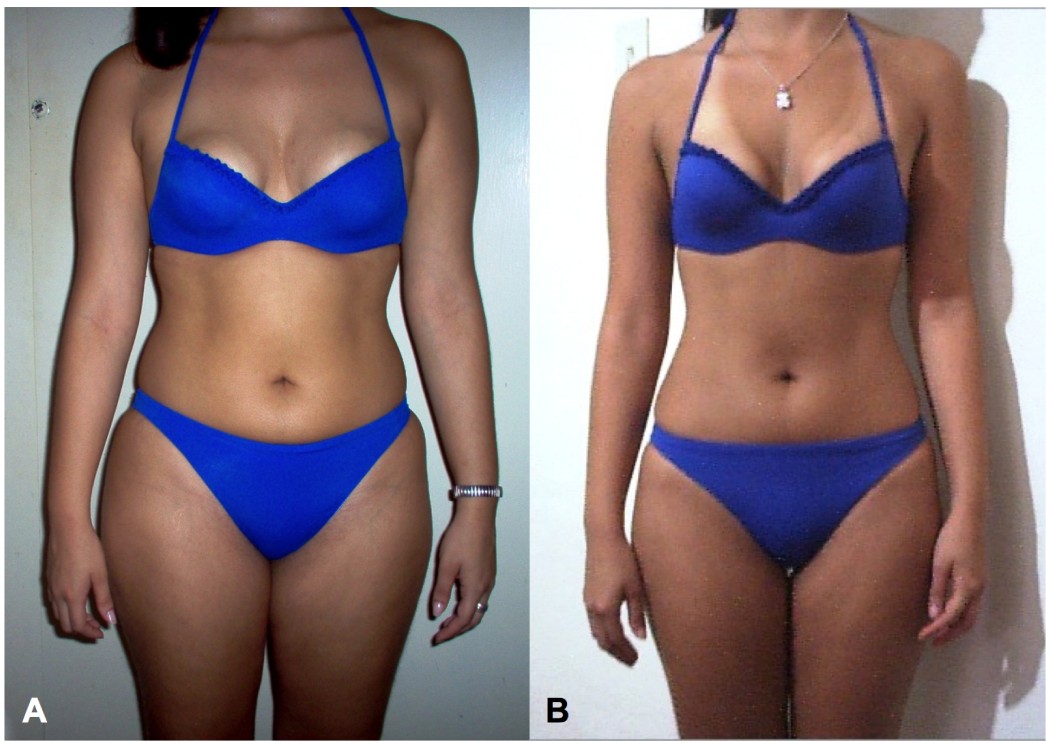

**Figure 4 Treatment photograph.** Patient's photographs at week 1 and week 16 help motivate the patient, demonstrating body image and self-perception changes. A: week number 1. B: week number 16.

16 treatment weeks, appear similar to waist loss in all BMI groups. Improved nutritional status is evident in the three BMI groups; all indicative body image parameters (waist, hip and waist/height ratio) significantly improved. We note that the waist/height ratio is an important parameter of nutritional improvement status and its relationship to health. Thus, this ratio is effective for predicting relative weight and simplifies the diagnosis of overweight and obesity (*Marrodán et al., 2011*).

## CONCLUSION

After three weeks of continuous treatment, patients improved all overweight and obesity parameters indicators; there were no statistically significant differences in hip circumference (HC) and waist loss (WC) (%) among the three BMI groups (normal-weight, overweight, and obesity). In contrast, there were statistically significant differences in weight loss (%) and waist-to-hip ratios. Based on anthropometric outcomes and patient perception of body image it can be concluded that the waist circumference loss is the parameter that retain obese patients in the weight loss program.

### Funding

The authors declare that there was no funding.

## Competing Interests

The authors declare there are no competing interests.

## Author Contributions

- Isaac Kuzmar conceived and designed the experiments, performed the experiments, wrote the paper, prepared figures and/or tables, reviewed drafts of the paper.
- Mercedes Rizo contributed reagents/materials/analysis tools, reviewed drafts of the paper.
- Ernesto Cortés-Castell analyzed the data, prepared figures and/or tables, reviewed drafts of the paper.

## Ethics

The following information was supplied relating to ethical approvals (i.e., approving body and any reference numbers):

SEMI-Servicios Médicos Integrados, Barranquilla, Colombia.

## Data Deposition

The following information was supplied regarding the deposition of related data:

Kuzmar, Isaac (2014): Obesity. Figshare.

http://dx.doi.org/10.6084/m9.figshare.1032566

## Supplemental Information

Supplemental information for this article can be found online at http://dx.doi.org/10.7717/peerj.495.

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
