# Peer review of "Adherence to an overweight and obesity treatment: how to motivate a patient?"

_PeerJ, doi:10.7717/peerj.495_

## Round 0.1 · original submission · Major Revisions

This is an interesting paper and suitable to be published in PeerJ, but not in its present form. It needs some ammendmends before being accepted.

Reviewer 1 ·

Basic reporting

It is a simple but very practical conclusion from a nutritional point of view and in order to motivate diet work, purpose often difficult for a large segment of the population.

Experimental design

I wanted to know how to select drew 16 weeks for the control time of the study.
Line 61: it is stated that a nutritional assessment was made, it is important to specify the type of dietary questionnaire used.
Line 63: it is stated that a nutritional program that was used. It is important to detail what was the program used and references of dyetetic recommendations also used.

Validity of the findings

in the study are evident significant differences in the ratio waist / hip before and after the regime? perhaps could include the data in the table before and after weight loss and comment on the text given the importance of this nutritional index

Reviewer 2 ·

Basic reporting

No Comments.

Experimental design

There is not shown a control group without WC measurement to affirm that "Patients do not dropout treatment because they start to see physical results in waist decrease". Self-perception data was also not included in the study to affirm this.

Validity of the findings

No Comments.

Additional comments

This paper is interesting and the results highlight the limitations of weight as a measurement in weight loss programs.
The manuscript is well written and structured. However, some major modifications should be made.

(1) Did the authors analysed the reasons why patients (n = 100) dropout or fail in the loss program? Did they not see physical results in waist circumference (WC) decrease? The manuscript would be benefit if the authors include this information.

(2) The aim of the study is “To determine the adhesion parameters in a treatment to improve body image in obese patients at a nutrition clinic”. However, the aim is more closely “To explore anthropometric changes in normal-weight, overweight and obese subjects who not dropout or failed a weight loss program over the 16 treatment weeks to improve patient’s motivation and treatment adherence”.

(3) The authors affirmed “Patients do not dropout treatment because they start to see physical results in waist decrease”. However, there is not shown a control group without WC measurement to affirm this. Moreover, there were not statistical significant differences in hip circumference (HC) and WC loss (%) among the three BMI groups (normal-weight, overweight, and obesity). In contrast, there were statistical significant differences in weight loss (%) and waist-to-hip ratio. Then, it can be concluded that it was WC changes that adhere obese patients to the weight loss program?

(4) The manuscript should be benefit if the authors could include data on body image and self-perception, as well as about motivation.

Some minor modifications should be also made in the manuscript:

Background
1. Line 6: “Body Mass Index (BMI, kg/m2), photographs (…)” instead of “Body Mass Index (BMI), photographs (…)”.
2. Line 11: “In 67 patients from 25≥BMI<30 group” instead of “In 67 patients from <25BMI<30 group”.
2. Line 12: “In 42 patients from BMI” instead of “42 patients from BMI”.
3. Line 15: “(…), with significant differences between the three groups (p<0.001)” instead of “(…), with significant differences between the three groups (p=0.000)”.

Introduction
1. Line 17: “Overweight (body mass index, BMI 25-30 kg/m2) and obesity (BMI ≥30 kg/m2) are preventable diseases defined as (…)” instead of “Overweight (BMI ≥25) and obesity (BMI ≥30) are preventable diseases defined as (…)”.
2. Line 38: “In morbid obesity (BMI ≥40 kg/m2)” instead of “In morbid obesity (BMI >40)”.

Material and methods
1. Some information has been repeated in “Subjects” and “Methods” sections. The manuscript should be benefit if the authors summarize these sections.

Results
1. The authors could also include the waist-to-height ratio parameter.

---

## Round 0.2 · accepted · Accept

The paper has been much improbed by the authors after they considered referees' suggestions. It is now suitable to be published in PeerJ.